# Validation of an Optical Technology for the Determination of pH in Milk during Yogurt Manufacture

**DOI:** 10.3390/foods13172766

**Published:** 2024-08-30

**Authors:** Siqi Liu, Fanny Contreras, Ricardo S. Alemán, Jhunior Marcía Fuentes, Oscar Arango, Manuel Castillo

**Affiliations:** 1Centre d’Innovació, Recerca i Transferència en Tecnologia dels Aliments (CIRTTA), Departament de Ciència Animal i dels Aliments, Facultat de Veterinària, Universitat Autònoma de Barcelona, 08193 Bellaterra, Spain; 2Department of Poultry Science, Auburn University, Auburn, AL 36849, USA; abigail.facz@gmail.com; 3School of Nutrition and Food Sciences, Louisiana State University Agricultural Center, Baton Rouge, LA 70802, USA; rsantosaleman@lsu.edu; 4Faculty of Technological Science, Universidad Nacional de Agricultura, Catacamas 16201, Honduras; jmarcia@unag.edu.hn; 5Faculty of Agroindustrial Engineering, Universidad de Nariño, Ciudad Universitaria Torobajo, Pasto 47154, Colombia; oscar.arango@udenar.edu.co

**Keywords:** yogurt fermentation, NIR light backscatter, optic sensor, inline, pH monitoring, temperature, protein concentration

## Abstract

Current systems that allow inline pH control in the fermented dairy industry have drawbacks, such as protein adhesion on the non-glass pH probes, measurement distortion, frequent recalibration needs, and sensitivity to extreme pH conditions encountered during clean-in-place operations. Therefore, the objective of this study was to validate the feasibility of estimating the pH of milk during the yogurt making process by using a NIR light backscatter sensor measuring under different fermentation temperatures and milk protein concentrations using a mathematical model that correlates the light scatter signal with pH. Three replications of the experiment with two protein concentrations (3.5 and 4.0%) and two fermentation temperatures (43 and 46 °C) were used to validate this inline pH prediction model. Continuous and discontinuous measurements of pH were collected as a reference during fermentation, simultaneously with the light backscatter data acquisition. Also, the effect of adjusting the initial voltage gain of the light scatter device on the accuracy of the pH prediction model was evaluated. Temperature and initial voltage were the main factors affecting the fitting accuracy of the model. The adjustment of the initial voltage gain improved the pH prediction model fit. The model has been successfully validated for both continuous and discontinuous measurements of pH, with SEP values < 0.09 pH units and CV < 1.78%. The proposed optical inline and non-destructive method was feasible for inline pH monitoring of milk fermentation, avoiding traditional manual pH measurement.

## 1. Introduction

Yogurt is one of the preferred dairy foods by consumers as a type of multi-functional food with high nutritional value, a relatively low price, and a long shelf life. Because of its huge market demand, the yogurt industrial production chain has matured, so the control of key points on the production line is required to be accurate, fast, and efficient. Milk fermentation is the most important stage in yogurt manufacturing [1]. Control of the milk fermentation process end point in industrial yogurt production is very important because of its relationship with product quality. If the pH is high, fermentation is incomplete, resulting in yogurt with an undesirable taste and texture. If the pH is too low, the yogurt may become too sour, and the texture may separate and release whey. Also, yogurt’s final pH in industrial production plants requires values lower than 4.6 due to food safety reasons.

To date, traditional electrochemical technology is the most common method for monitoring the fermentation process and determining the pH of fermented dairy products. This measurement method is usually carried out in a discontinuous manner because continuous measurement can cause a series of problems, such as protein adhesion on the non-glass pH probes, measurement distortion [2], frequent recalibration needs, and sensitivity to extreme clean-in-place pH conditions [3]. However, manually collecting samples every 10–15 min may lead to contamination of milk batches and poor real-time performance.

A fiber optic sensor technology used to measure light dispersion has been demonstrated to be a convenient, inline, and nondestructive method for monitoring milk coagulation. Light from a light-emitting diode (LED) is transferred to the milk through an optic fiber, and the light backscattered (LB) from the milk is transmitted through an adjacent fiber to an optical detector. The LB signal contains information about the aggregation of casein micelles and gel assembly during rennet coagulation [4,5,6].

Arango et al. (2020a) [3] evaluated the suitability of an optical sensor as a new method for the inline control of yogurt fermentation. Tests were conducted with three fat and three inulin concentrations, and fermentation was monitored simultaneously using an inline light backscatter sensor, pH meters, and a rheometer. A mathematical model that correlates the near-infrared light backscatter ratio with the pH in real time during milk fermentation was developed. The model was calibrated and successfully validated under the different experimental conditions.

The proposed optical technology for the inline control of milk fermentation, whose intellectual property belongs to the Universitat Autònoma de Barcelona (UAB), combines the use of a near-infrared (NIR; 880 nm) light-backscatter fiber optic sensor with a specific algorithm to convert, in real time, the sensor response into pH estimations. This technology can be operated inline, does not invade or destroy the sample, meets hygienic requirements, does not require continuous maintenance or pH calibration after installation, and does not consume material or reagents. It avoids the drawbacks of traditional technology, can better determine the optimal incubation time, and complies with food regulations while reducing operating costs.

The objective of this study was to validate the feasibility of estimating the pH of milk during the yogurt making process by a NIR light backscatter sensor under different fermentation temperatures and milk protein concentrations using the mathematical model proposed by Arango (2020a) [3], because in this study only fat and inulin concentrations were evaluated and it was necessary to validate if the method and model work properly under other production conditions.

## 2. Materials and Methods

An experiment was designed in which yogurts were made from milk with different protein contents and with two fermentation temperatures. In each test, the variation of the pH and the NIR light backscatter ratio (R) were measured, in order to obtain and validate models that transform R data into pH measurements.

### 2.1. Experimental Design

An experiment with three replications was designed and performed to study the effects of two different levels of incubation temperature (43 and 46 °C) and protein concentration (3.5 and 4.0%) on the yogurt fermentation process. These values were chosen because they are commonly used in the dairy industry. The design had a total of 12 tests, which were performed randomly.

The light backscatter unit used in this experiment is equipped with two measuring vats prepared to continuously measure milk pH and light backscatter in parallel during coagulation. In order to evaluate if the effect of protein concentration and temperature on the initial voltage can be corrected and its effect on the pH prediction models used to improve the prediction algorithm performance, a correction on the initial voltage in one of the vats (vat #1) was made for each test, while in the other vat (vat #2), it was allowed to vary freely. The mentioned voltage correction is explained in detail in Section 2.5.2.

In order to evaluate the pH progress in a similar manner to that used during industrial yogurt manufacturing, an aliquot of the inoculated milk was coagulated in a beaker inside a water bath at the same target temperature used in the light scatter unit, and samples were obtained every ~8 min to evaluate the pH progress using a regular external pH meter. Then, continuous and discontinuous pH measurements were correlated to light scatter readings to pursue validation of the optical pH prediction method at different protein concentrations and fermentation temperatures.

### 2.2. Preparation of Milk

This experiment used commercially skimmed UHT milk purchased from a local supermarket in Spain as a raw material. The same batch of milk was used for the whole experiment in order to minimize the variability associated with uncontrolled milk composition or pretreatment.

The 1 L bricks of milk were kept at 4 °C until opened and used. The milk sample was adjusted to the protein concentration required for each test using low-heat skim milk powder (Chr. Hansen, Barcelona, Spain). The protein and fat composition of both skim UHT milk and skim milk powder is shown in Table 1.

In this experiment, samples of ~500 mL containing the target concentration of protein, according to the experimental design, were prepared by mixing skimmed UHT milk and skimmed milk powder. For each test, the sample was prepared as follows: 500 mL of skimmed UHT milk was measured in a volumetric flask, and the amount of milk powder required to achieve the target protein concentration was calculated and weighed in a beaker and added to the UHT milk. After this, the mixture was stirred with a magnetic stirrer for 30 min at 43 °C and then left to stand in the dark for another 30 min, protecting the sample from the air using parafilm. The sample was heated to 90 °C and left at this temperature for 5 min, after which it was quickly cooled to the target fermentation temperature, 43 °C or 46 °C, using an ice bath.

### 2.3. Preparation of Starter Culture

Considering the high content of live bacteria in direct-vat-starter cultures (DVS), they are convenient and quick to use, simplifying the fermentation process. Furthermore, the use of this type of starter culture has a lower risk of phage infection during the fermentation process. This experiment used the commercially lyophilized culture of *Streptococcus thermophilus* and *Lactobacillus delbrueckii* subsp*. Bulgaricus* (YO-MIX 496 LYO 100 DCU, Dasnisco, Sassenage, France) as a starter culture for yogurt fermentation.

With the aim of maximizing the activity of the inoculum, the commercial culture was first grown in skim milk. On the day of each test, 88 mL of skimmed milk powder, rehydrated to 12% total solids, was stirred and heat-treated at 90 °C for 5 min, as described in point 2.2. Then, it was cooled to 43 °C, inoculated with 130 mg L^−1^ DVS, and the mixture was stirred and incubated at 43 °C until pH 5.0 was reached [7]. A total of 2% of this working culture was used as an inoculum for subsequent fermentation of the test milk sample.

### 2.4. Acid Milk Coagulation Induction

For each test, the amount of skimmed milk powder needed was calculated based on the protein content required for the experiment and added to 500 mL of UHT milk as described in Section 2.2. The protein-adjusted mixture of UHT and milk powder was used for testing acid coagulation, as shown in Figure 1.

The temperature control system was previously set at its required incubation temperature (43 °C or 46 °C). Then the mixture was left in a thermostatic bath at the corresponding incubation temperature until thermal equilibrium was reached. At that time, 10 g (2%) of working culture prepared as described in Section 2.3 was added, and the whole liquid was stirred with a spatula for 1 min. Two aliquots of 80 mL were poured into the two vats of the optical unit, which will be described in Section 2.5.1. At each vat, pH electrodes were placed through a hole located in the lid of the vat. Data acquisition corresponding to both light backscatter sensors and pH electrodes (vats #1 and #2) was immediately initiated at the time of inoculum addition. The remaining sample (340 mL) was placed in a sealable beaker and sampled every 8 min to measure the pH using a standard pH meter connected to a glass pH electrode.

### 2.5. Measurement of the Light Backscatter Ratio and pH

#### 2.5.1. Determination of NIR Light Scatter Parameters

The optical apparatus used to determine near-infrared light backscatter at 880 nm during milk coagulation, named CoAguLab, was designed at the University of Kentucky. Its design was described in detail in the paper of Tabayehnejad et al. (2012) [5]. A brief description follows. It has two vats that simultaneously monitor the acid coagulation of two samples for accurate comparison (Figure 2). An optic unit directs near-infrared light from an LED emitting at 880 nm to the milk sample through an optical fiber, while a second fiber returns backscattered light at 180 degrees back to a silicon detector. The wavelength of 880 nm has been used repeatedly in several works where the NIR light scattering sensor has been used to evaluate the enzymatic and acid coagulation of milk because it is the one that best responds to micellar aggregation processes [7,8].

The scattered light is linearly converted by the sensor into a voltage signal. The voltage is measured every two seconds, and the average of three measurements is recorded every six seconds. The equipment is zeroed by switching off the LED and adjusting the voltage reading to zero volts. Once milk coagulation monitoring is initiated, the first ten voltage registers are averaged to calculate the initial voltage (V0). Once V0 is calculated, the light-backscatter ratio (R) is obtained by dividing the voltage measured every six seconds by V0.

#### 2.5.2. Adjustment of the Voltage Gain

To evaluate if the effect of protein concentration and temperature on the initial voltage could be corrected, the following procedure was carried out: before starting data acquisition, the voltage of vat #1 was adjusted to 2.00 V, using the sample prepared for each test, when the treatment temperature was in equilibrium. Contrarily, for vat #2, the voltage was only adjusted to 2.00 V before the first treatment of each replication, and then it was allowed to vary freely according to coagulation temperature and protein concentration in each test.

#### 2.5.3. pH Measurement

The development of the proposed optical pH prediction model was done based on continuous and simultaneous acquisition of both pH and light backscatter measurements during acid coagulation as a function of time. However, the yogurt fermentation end-point is selected worldwide by sampling yogurt every 10–15 min until pH 4.6 is reached. As a result, two pH measurement procedures were used in this study. On one hand, the pH of each vat (#1 and #2) was measured using separate pH electrodes placed on the milk of the vat through a hole in the lid of each vat. The electrodes (Thermo Scientific™ Orion 8104BN ROSS, Basel, Switzerland) were connected to the data acquisition enclosure of the CoAguLab tester. These pH measurements were collected every 6 s. On the other hand, the milk sample that was fermented inside the external water bath, in parallel to those samples placed in vats #1 and #2, was sampled manually. Every 8 min, a small aliquot was collected and placed in a small beaker. The sample was stirred, and the pH was measured using an external pH electrode (Model 50 12T, Crison Instruments, S.A., Barcelona, Spain) connected to a pH meter (Model pH BASIC 20, Crison Instrument). As the discontinuous pH data was taken every 8 min, the external pH curves were adjusted as a function of time by polynomial expressions in order to estimate pH data every 6 s. This procedure allowed us to expand the number of datapoints for calibration.

#### 2.5.4. pH Electrode Calibration

Prior to each test, the electrodes attached to the CoAguLab system and the one connected to the external pH meter were calibrated separately using the standard buffer solutions of pH = 7.00 and pH = 4.01 at the corresponding tested temperatures. After calibration was complete, pH electrodes were stored in the storage solutions recommended by the manufacturers.

While electrodes connected to the CoAguLab unit were stored using storage solution Cat. No. 810001, the external electrode used storage solution CRISOLYT-A (KCl 3M + AgCl). When each replication was completed, all the electrodes were cleaned, following the manufacturer’s cleaning protocol and the recommended cleaning solution, to prevent protein precipitation and salt deposits.

#### 2.5.5. Statistical Analysis

A prediction model that transforms the light backscatter ratio measured by the acid coagulation tester into real-time pH measurements (pH = f (R)), proposed by Arango (2020a) [3], was calibrated and validated in this study. As established by this author, the row data corresponding to the pH values in the range of 5.2–4.6 were selected as the “working” data set for statistical analysis.

Two of the three replications were used for the calibration of the model. Three possible two-replication combinations were tested for calibration: replications one-two, one-three, and two-three, while the replication not employed for calibration in each of the three cases was utilized for validation. Calibration of the model was performed using CurveExpert software (CurveExpert Professional version 2.6.5, Daniel G. Hyams, Huntsville, AL, USA), which allowed for the estimation of the four different regression coefficients contained in the prediction model.

In addition, different adjustments of the initial voltage were made, as explained in detail in Section 2.5.2, in order to evaluate statistically which of the two procedures allows a better adjustment of the pH prediction model.

## 3. Results and Discussion

In this study, lactic acid fermentation was carried out under different fermentation temperatures and protein concentration conditions. The relationship between the light backscatter ratio (R), the pH profiles, and the first derivative of R as a function of time is shown in Figure 3 and Figure 4. Since the curves obtained under the same protein concentration and temperature for each of the three replications were approximately the same, the data in Figure 3 and Figure 4 was selected from replica 2.

As indicated by Figure 3 and Figure 4, milk pH before yogurt fermentation was ~6.5–6.4. Bacterial growth initiated a slow decrease in pH induced by lactic acid production from lactose. The coagulation of yogurt happened in two stages, and the first stage was defined by calculating the t_max_ value obtained by the first derivative of R vs. time (Figure 3a,b and Figure 4a,b). The decline in pH was at its maximum rate when pH reached a value of ~5.7–5.5, corresponding to t_max_, where a first aggregation occurred due to the denatured particles of the serum proteins that bind to each other and with the casein micelles. This was consistent with the results of both Arango (2020b) [7] and Lee and Lucey (2004) [10]. Then, there was a second stage, which was identified by the second maximum of R (t_max2_). During the fermentation process, the decrease in milk pH caused the colloidal calcium phosphate (CCP) within casein micelles to solubilize. This process is typically completed when the pH is ~5.0, if most whey proteins remain native [11]. At this point, the milk pH is close to the isoelectric point of casein (IP~4.6), which helps to enhance the attraction between the casein and thus increases the gel hardness [10,11]. However, applying an intense heat treatment to milk prior to fermentation denatures a significant amount of whey proteins, which attach to the surface of casein micelles, inducing an IP shift as a result of the higher isoelectric pH of whey proteins [12].

Thus, due to the heat treatment of milk applied in this experiment (90 °C during 5 min), denaturation of whey proteins modified the IP at which acid coagulation occurred. Based on the second maximum of the first derivative in Figure 3a,b and Figure 4a,b, the beginning of the second stage of milk fermentation, the aggregation of demineralized and destabilized micelles, took place at a pH~5.2. From that moment on, the gel hardening continues as the pH continues to decrease.

Figure 3a and Figure 4a showed the first derivative curve (R’) of R with different protein concentrations at 43 °C, while, similarly, Figure 3b and Figure 4b showed the first derivative curve of R at 46 °C. Comparing Figure 3a with 3.5% protein concentration and Figure 4a with 4.0% protein concentration (both at 43 °C), it was found that different protein concentrations had no effect on the values of t_max_ and t_max2_. Similar behavior was observed for the effect of protein at 46 °C (Figure 3b and Figure 4b). Conversely, Figure 4a,b (4.0% protein but different temperature) showed that temperature may affect the rate of coagulation, which was expected.

When the concentration of protein remained unchanged at 4.0%, the first stage of aggregation was relatively late at 43 °C (Figure 4a), at which the value of t_max_ was 93.2 min; and when the temperature was raised from 43 °C to 46 °C, microbial metabolism and physicochemical reactions were more accelerated, which made the aggregation of the first stage of fermentation quicker, and t_max_ advanced to 81.4 min (Figure 4b). Similarly, the onset of coagulation and hardening, demarcated by t_max2_, may be anticipated with the increase in temperature. Comparing Figure 4a,b, at the same protein concentration (4.0%), the value of t_max2_ corresponding to 43 °C was 110.4 min, while when the fermentation temperature was raised to 46 °C, t_max2_ was 12.7 min shorter (97.7 min). Similar behavior as a function of temperature was observed for t_max_ and t_max2_ at 3.5% protein (Figure 3a,b).

Various studies have shown that increasing the fermentation temperature increased whey separation, which was the same as the experimental phenomenon observed in this experiment, namely, that the yogurt fermented at 46 °C produced more whey (not measured but observed). The papers of Lucey (2001) [13] and Melema et al. (2002) [14] demonstrated that high incubation temperatures made the gel network more unstable and were more prone to protein network rearrangement, resulting in greater whey separation.

The experimental results corresponding to calibration and validation of the optical pH prediction model for the three different pH acquisition systems evaluated were analyzed separately and are presented below [9].

### 3.1. Results of the Model without Voltage Gain Adjustment (Vat #2)

#### Calibration and Validation

Calibration and validation of the pH prediction model were performed using experimental data corresponding to each temperature and protein concentration combination, according to the method described previously (Section 2.5.5). The resulting coefficients of determination (R^2^), standard errors of prediction (SEP), and coefficients of variation (CV) for model calibration as well as validation were used as model performance indicators and are shown in Table 2.

It was evident from Table 2 that model calibration and validation were greatly affected by temperature. For a protein concentration of 3.5%, increasing the fermentation temperature from 43 to 46 °C resulted in a decrease in the model performance indicators. The accuracy of the pH prediction model is indicated by the high value of R^2^ and the small values of both SEP and CV [15]. Also, the observed effect of fermentation temperature on model calibration and validation for a protein concentration of 4.0% was quite similar to that of 3.5% protein [16]. So as the temperature rose, the results showed that the fitting accuracy was lower. On the other hand, it was observed that increasing the protein concentration worsened the model adjustment, although the result is less clear than the effect of increasing the temperature.

For treatment combinations A, B, C, and D, the data were the means ± standard deviations of three replications. R^2^c, coefficient of determination of calibration; R^2^v, coefficient of determination of validation; SEPc, standard error of prediction of calibration (pH units); SEPv, standard error of prediction of validation (pH units); CVc, coefficient of variation of calibration (%); CVv, coefficient of variation of validation (%). N, number of experiments; N = 12. Nc, the total number of calibration datapoints, is 3450. Nv, the total number of validation datapoints; is 1725.

Moreover, from the results in Table 2, the average value of R^2^v was only 0.827 ± 0.135, well below the value corresponding to treatment combination A (3.5% protein and 43 °C; R^2^v = 0.989 ± 0.003). Thus, with unadjusted voltage gain, increasing protein and temperature levels may affect the accuracy of the prediction models.

The relationship between the predicted and observed pH values is shown in Figure 5.

Based on Figure 5, it should be highlighted that in all four evaluated conditions, SEP was <0.094 pH units, with CV < 2%. However, it was evident from the distribution of the residuals along the pH scale that only the results at 3.5% protein concentration and 43 °C were in line with expectations and showed the best fit between real and predicted pH data (Figure 5a), while the model fit was worse for fermentation at 46 °C (Figure 5c,d). These results suggested that the absence of an initial voltage gain adjustment seems to negatively affect the accuracy of the pH prediction model.

### 3.2. Results of the Model with Voltage Gain Adjustment (Vat #1)

#### Calibration and Validation

Similarly to the methodology described in Section 3.1., model calibration and validation were performed on the data obtained at the same protein concentration and temperature, and the results corresponding to model performance indicators are shown in Table 3.

For treatment combinations A, B, C, and D, the data were the means ± standard deviations of three replications. R^2^c, coefficient of determination of calibration; R^2^v, coefficient of determination of validation; SEPc, standard error of prediction of calibration (pH units); SEPv, standard error of prediction of validation (pH units); CVc, coefficient of variation of calibration (%); CVv, coefficient of variation of validation (%). N, number of experiments; N = 12. Nc, the total number of calibration datapoints, is 3396. Nv, the total number of validation datapoints, is 1698.

According to the results shown in Table 2 and Table 3, the model with adjusted initial voltage gain yielded better predictions than those of vat #2 (without adjustment). According to the inference in Section 3.1, when protein remained the same, the higher the temperature, the lower the fitting accuracy of the prediction model. Similarly, with the voltage gain adjusted, when the protein concentration was 3.5%, the accuracy of the prediction model was reduced when the temperature increased from 43 °C to 46 °C. This same effect also applied to 4.0% protein concentration. On the other hand, similar values of R^2^v and SEPv were obtained at 43 °C for both protein levels. This result also happened at 46 °C. The best R^2^v, SEPv, and CVv values for validation were obtained at a combination of 43 °C and 3.5% protein concentration.

In order to show the data from Table 2 more intuitively, replication 2 with the highest predicted pH accuracy in the model was selected, and the relationship between the pH predicted value and the true value is shown in Figure 6.

From Figure 6a,b, it was evident that the degree of coincidence between the pH predicted and true values was quite high, especially at 43 °C. Although the accuracy of Figure 6c,d was not completely satisfying, the predictions were better than the results of Figure 5c,d from vat #2.

### 3.3. Results of the External pH Model with Voltage Gain Adjustment (pHE)

It should be noted that the pH values obtained by this method were discontinuous, and the light backscatter ratio (R) used was acquired from vat #1 of CoAguLab equipment, as its performance was clearly better.

#### Calibration and Validation

Following the same procedure discussed in Section 3.1 and Section 3.2, the average values of the three replications obtained at different temperatures and protein concentrations are shown in Table 4.

For treatment combinations A, B, C, and D, the data were the means ± standard deviations of three replications. R^2^c, coefficient of determination of calibration; R^2^v, coefficient of determination of validation; SEPc, standard error of prediction of calibration (pH units); SEPv, standard error of prediction of validation (pH units); CVc, coefficient of variation of calibration (%); CVv, coefficient of variation of validation (%). N, number of experiments; N = 12. Nc, the total number of calibration datapoints, is 4290. Nv, the total number of validation datapoints, is 2145.

Although the light backscatter ratio (R) used in this modeling method was the same as that used in Section 3.2, there were differences in the pH profiles as they were measured externally in an attempt to reproduce the current industrial pH determination procedure. It can be observed that the average values of each validation performance indicator obtained for Section 3.2 (Table 3) were better than those of Table 4. The effect of temperature and protein on the prediction coefficients was similar to those already discussed in Section 3.1 and Section 3.2. As in previous cases (vats #1 and #2), the fitting of the established prediction model was optimal under conditions of 3.5% protein concentration and a fermentation temperature of 43 °C. The relationship between pH predicted (for pH values obtained in a discontinuous way) and true values is shown in Figure 7.

The results of Figure 7a,b showed that the accuracy of the pH prediction model for pH data taken off-line with an external pH electrode was optimal at 43 °C, in line with expectations. However, at 46 °C, although the pH predicted values shown in Figure 7c,d were not suitable, the predictions were better than those shown in Figure 5c,d corresponding to vat #2 (no voltage gain adjustment) under the same experimental conditions.

As observed in Figure 5, Figure 6 and Figure 7, it can be concluded that in all cases, the pH prediction model showed a better fit at lower protein levels and fermentation temperatures. The adjustment in the voltage gain allowed for improved pH prediction at both low and high protein levels and temperatures, even for the pH data obtained discontinuously with an external electrode.

Therefore, the optical sensor in combination with the prediction model could be used for inline monitoring of the yogurt fermentation process, and the highest fitting accuracy was obtained in the equipment with adjusting initial voltages when the acidification temperature was set at 43 °C and the pH range was below 5.2.

## 4. Conclusions

The results showed that the model for inline prediction of pH during milk fermentation using light backscatter data was successfully validated at different milk protein concentrations and fermentation temperatures. The best model fit for pH prediction was obtained in treatments with 3.5% protein and a 43 °C fermentation temperature. Increasing fermentation temperature from 43 to 46 °C lowered the model fitting accuracy; however, the R^2^ and SEP values were not affected by increasing milk protein concentration. Adjusting initial voltages improved the pH prediction model fitting, increasing R^2^ from 0.989 to 0.998 for treatments with 3.5% protein at 43 °C. The proposed optical inline and non-destructive method was feasible for inline pH monitoring of milk fermentation, avoiding traditional manual pH measurement. Further validation of the method is necessary to improve the fit of pH prediction at high protein levels or fermentation temperatures. It should also be validated under other yogurt production conditions that have not yet been studied.

## Figures and Tables

**Figure 1 foods-13-02766-f001:**
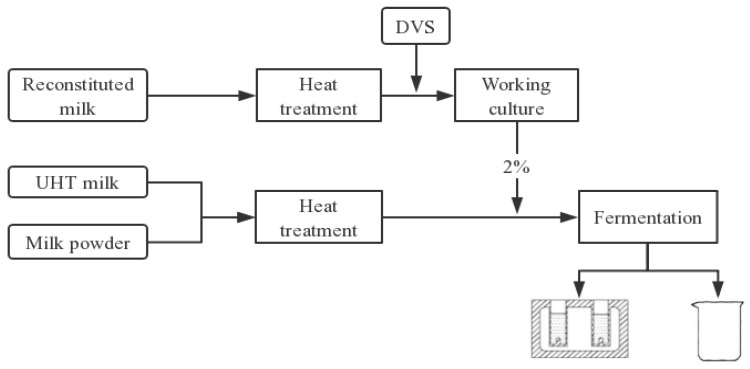
A schematic diagram for comparing the traditional technique for yogurt fermentation end-point selection with the alternative optical end-point selection method using near-infrared light backscatter.

**Figure 2 foods-13-02766-f002:**
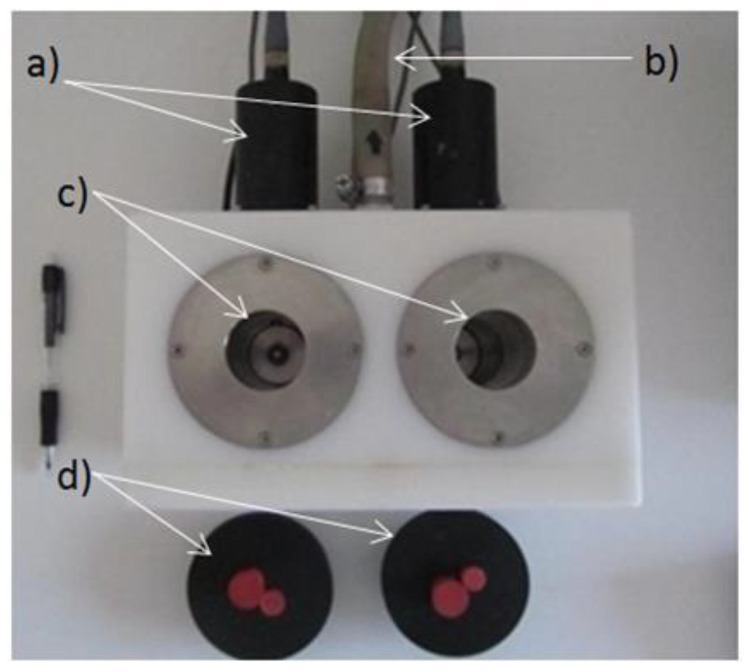
Top view of the CoAguLab unit showing: (a) optical sensors; (b) water outlet for temperature control (the water inlet is below); (c) stainless steel vats where the samples are deposited; (d) plastic caps to prevent surface evaporation, with access to insert pH electrodes [9].

**Figure 3 foods-13-02766-f003:**
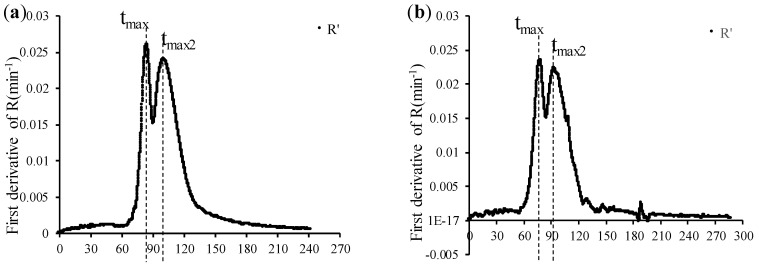
The relationship between the light backscatter ratio and pH profiles with 3.5% protein concentration at (**c**) 43 °C and (**d**) 46 °C. The first derivative of R versus time is at (**a**) 43 °C and (**b**) 46 °C. Data correspond to vat #1 of replica 2. R, light backscatter ratio; pH, pH value measured by CoAguLab; pH_E_, discontinuous, external pH measurements. R’, first derivative of R (min^−1^); t_max_, first maximum of the first derivative; t_max2_, second maximum of the first derivative.

**Figure 4 foods-13-02766-f004:**
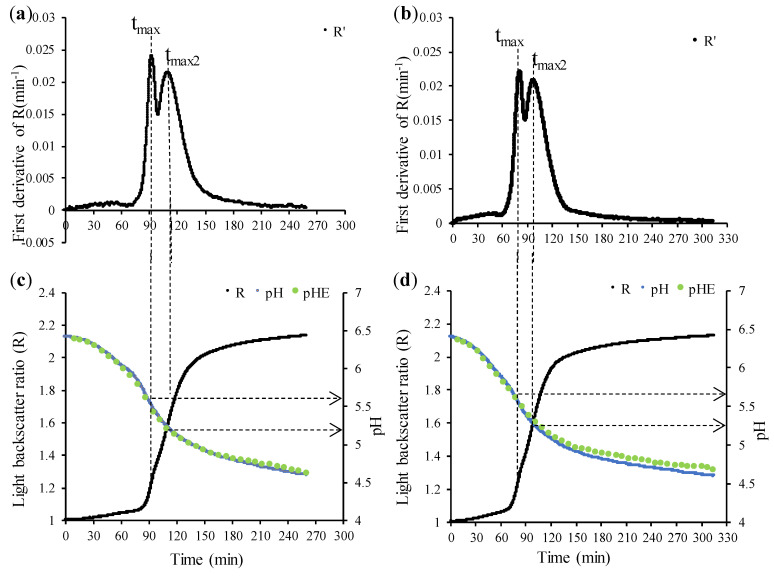
The relationship between the light backscatter ratio and pH profiles with 4.0% protein concentration at (**c**) 43 °C and (**d**) 46 °C. The first derivative of R versus time is at (**a**) 43 °C and (**b**) 46 °C. Data correspond to vat #1 of replica 2. R, light backscatter ratio; pH, pH value measured by CoAguLab; pH_E_, discontinuous, external pH measurements; R’, first derivative of R (min^−1^); t_max_, first maximum of the first derivative; t_max2_, second maximum of the first derivative.

**Figure 5 foods-13-02766-f005:**
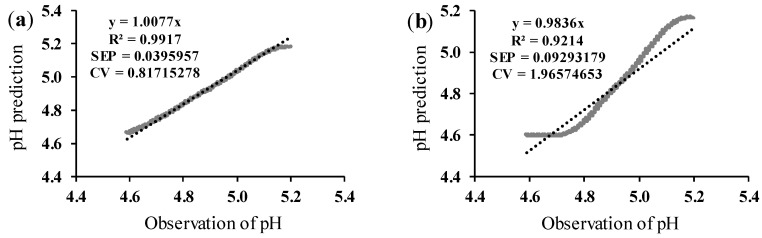
Validation of the pH prediction model without initial voltage gain adjustment. Validation data correspond to replication 3. N, number of validation datapoints; R^2^, coefficient of determination; SEP, standard error of prediction (pH units); CV, coefficient of variation (%); (**a**) data at 43 °C and 3.5% of protein, N = 1366; (**b**) data at 43 °C and 4.0% of protein, N = 1516; (**c**) data at 46 °C and 3.5% of protein, N = 2067; (**d**) data at 46 °C and 4.0% of protein, N = 2032.

**Figure 6 foods-13-02766-f006:**
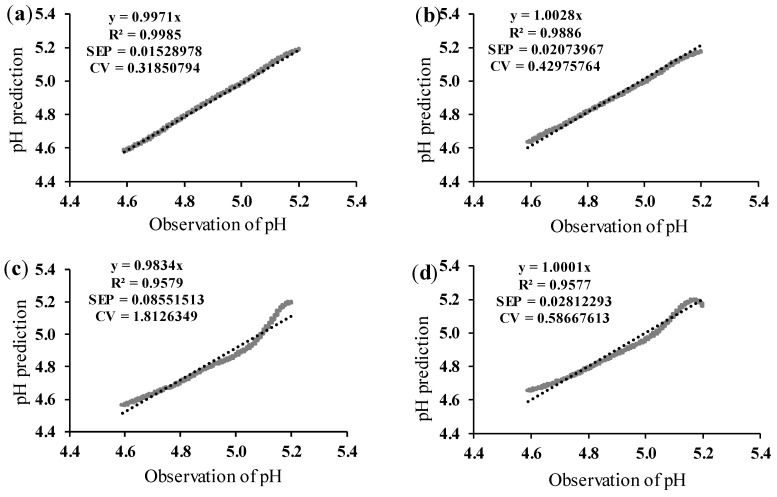
Validation of the pH prediction model, with adjusted initial voltage gain. Validation data correspond to replication 2. N, number of validation datapoints; R^2^, coefficient of determination; SEP, standard error of prediction (pH units); CV, coefficient of variation (%). (**a**) Data at 43 °C and 3.5% of protein. N = 1382; (**b**) data at 43 °C and 4.0% of protein. N = 1480; (**c**) data at 46 °C and 3.5% of protein. N = 1856; (**d**) data at 46 °C and 4.0% of protein. N = 2095.

**Figure 7 foods-13-02766-f007:**
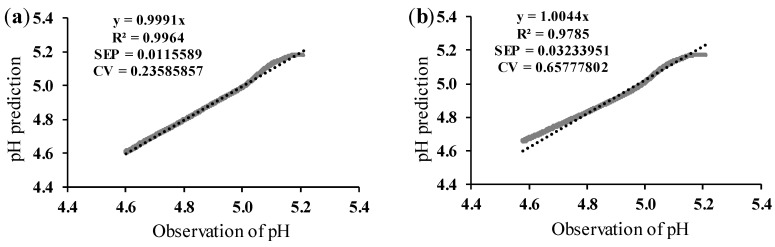
Validation of the pH prediction model, with adjusted initial voltage gain for pH data from the external pH electrode. Validation data corresponding to replication 2. N, number of validation datapoints; R^2^, coefficient of determination; SEP, standard error of prediction (pH units); CV, coefficient of variation (%). (**a**) Data at 43 °C and 3.5% of protein. N = 1742; (**b**) data at 43 °C and 4.0% of protein. N = 1857; (**c**) data at 46 °C and 3.5% of protein. N = 2503; (**d**) data at 46 °C and 4.0% of protein. N = 2571.

**Table 1 foods-13-02766-t001:** Concentration of protein and fat in the raw materials used in the experiment ^a^.

Concentration	Skimmed UHT Milk (g/100 mL)	Skimmed Milk Powder (g/100 g)
Protein (%)	3.2	36.5
Fat (%)	0.3	0.9

^a^ Information taken from the product label.

**Table 2 foods-13-02766-t002:** Model performance indicators obtained at different temperature and protein concentration levels (vat #2).

Treatments		A		B		C		D		Average ± SD
Parameters		T(°C)	P (%)	T (°C)	P (%)	T (°C)	P (%)	T (°C)	P (%)	
		43	3.5	43	4.0	46	3.5	46	4.0	
**Calibration**	**R^2^_c_**	0.972 ± 0.023		0.855 ± 0.089		0.748 ± 0.185		0.792 ± 0.097		0.792 ± 0.097
	**SEP_c_**	0.024 ± 0.014		0.054 ± 0.015		0.065 ± 0.022		0.062 ± 0.012		0.051 ± 0.016
	**CV_c_**	0.493 ± 0.297		1.126 ± 0.321		1.365 ± 0.455		1.300 ± 0.244		1.071 ± 0.329
**Validation**	**R^2^_v_**	0.989 ± 0.003		0.833 ± 0.133		0.771 ± 0.147		0.715 ± 0.258		0.827 ± 0.135
	**SEP_v_**	0.046 ± 0.024		0.085 ± 0.036		0.116 ± 0.048		0.107 ± 0.054		0.089 ± 0.040
	**CV_v_**	0.965 ± 0.504		1.769 ± 0.728		2.400 ± 0.949		2.267 ± 1.191		1.850 ± 0.843

**Table 3 foods-13-02766-t003:** Model performance indicators obtained at different temperature and protein concentration levels (vat #1).

Treatments		A		B		C		D		Average ± SD
Parameters		T (°C)	P (%)	T (°C)	P (%)	T (°C)	P (%)	T (°C)	P (%)	
		43	3.5	43	4.0	46	3.5	46	4.0	
**Calibration**	**R^2^_c_**	0.998 ± 0.001		0.988 ± 0.009		0.877 ± 0.099		0.916 ± 0.062		0.945 ± 0.043
	**SEP_c_**	0.008 ± 0.002		0.017 ± 0.007		0.048 ± 0.022		0.041 ± 0.014		0.028 ± 0.011
	**CV_c_**	0.166 ± 0.039		0.344 ± 0.153		0.994 ± 0.457		0.860 ± 0.291		0.591 ± 0.235
**Validation**	**R^2^_v_**	0.998 ± 0.001		0.994 ± 0.005		0.932 ± 0.023		0.943 ± 0.039		0.967 ± 0.017
	**SEP_v_**	0.012 ± 0.005		0.030 ± 0.012		0.086 ± 0.042		0.070 ± 0.036		0.049 ± 0.023
	**CV_v_**	0.240 ± 0.094		0.617 ± 0.248		1.775 ± 0.834		1.454 ± 0.755		1.021 ± 0.483

**Table 4 foods-13-02766-t004:** Model performance indicators obtained at different temperature and protein concentration levels with pH data obtained with an external pH electrode.

Treatments		A		B		C		D		Average ± SD
Parameters		T (°C)	P (%)	T (°C)	P (%)	T (°C)	P (%)	T (°C)	P (%)	
		43	3.5	43	4.0	46	3.5	46	4.0	
**Calibration**	**R^2^_c_**	0.993 ± 0.004		0.975 ± 0.018		0.878 ± 0.087		0.862 ± 0.088		0.927 ± 0.049
	**SEP_c_**	0.015 ± 0.005		0.026 ± 0.012		0.050 ± 0.021		0.053 ± 0.015		0.036 ± 0.013
	**CV_c_**	0.302 ± 0.096		0.526 ± 0.250		1.014 ± 0.435		0.973 ± 0.012		0.731 ± 0.271
**Validation**	**R^2^_v_**	0.994 ± 0.003		0.973 ± 0.012		0.750 ± 0.323		0.788 ± 0.137		0.876 ± 0.119
	**SEP_v_**	0.022 ± 0.010		0.049 ± 0.025		0.081 ± 0.035		0.102 ± 0.056		0.064 ± 0.031
	**CV_v_**	0.459 ± 0.200		1.009 ± 0.515		1.632 ± 0.682		2.093 ± 1.176		1.298 ± 0.643

## Data Availability

The original contributions presented in the study are included in the article, further inquiries can be directed to the corresponding author.

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
