# Peer review of "Validation of an Optical Technology for the Determination of pH in Milk during Yogurt Manufacture"

_foods, 2024, doi:10.3390/foods13172766_

Round 1
Reviewer 1 Report
Comments and Suggestions for Authors
The manuscript represents a contribution to the field of in-line pH monitoring in yogurt fermentation. The proposed optical method, using NIR light backscattering and a mathematical model, has the potential to overcome the limitations of traditional approaches and improve the efficiency and accuracy of yogurt production to improve. The performance of the model appears to be less robust at higher fermentation temperatures, which may require further optimization or exploration of alternative models. The study focuses primarily on laboratory-controlled conditions, and a direct comparison with industrial yogurt production practices would further strengthen the impact of the manuscript.
· Introduction – While the introduction discusses the importance of pH monitoring in yogurt production, the validation aspect contained in the title is not explicitly mentioned. The first paragraph could be more specific about the connection between accurate pH monitoring and yogurt quality. The introduction relies heavily on the limitations of traditional methods. Although it is important to focus earlier on the proposed optical technology. In the second paragraph, introduce the concept of NIR backscattering and its potential benefits. Explicitly state the connection between the validation of NIR technology and its application in yogurt production. In this context, mention the model of Arango (2005)..
· Materials and Methods – Consider adding a short introduction to this section that summarizes the entire experimental approach. In Section 2.2, explain why the same batch of milk was used throughout the experiment. In Section 2.5.1, you might consider mentioning the reasons for choosing 880 nm as the wavelength for NIR light.
· Results and discussion – In Section 3.1, consider consolidating results and discussions. Report the observed influence of temperature and protein concentration on model performance without going into too much detail about specific values. Similarly, Sections 3.2 and 3.3 combine results and discussion, focusing on the key findings. Highlight the improvement in model performance through voltage gain adjustment and optimal conditions (3.5% protein, 43°C). Describe the key features of Figures 5, 6, and 7 in the text, emphasizing the trends and influence of temperature and protein concentration.
· Conclusions – Quantify the improvement achieved by voltage gain adjustment. Note the limitations of the model at higher fermentation temperatures. Mention potential areas for further research. Simplify some sentences for better readability.
· Abstract – The summary can be shortened by removing unnecessary details. Emphasize the advantages of the proposed method compared to traditional approaches.
Author Response
The manuscript represents a contribution to the field of in-line pH monitoring in yogurt fermentation. The proposed optical method, using NIR light backscattering and a mathematical model, has the potential to overcome the limitations of traditional approaches and improve the efficiency and accuracy of yogurt production to improve. The performance of the model appears to be less robust at higher fermentation temperatures, which may require further optimization or exploration of alternative models. The study focuses primarily on laboratory-controlled conditions, and a direct comparison with industrial yogurt production practices would further strengthen the impact of the manuscript.
- Introduction – While the introduction discusses the importance of pH monitoring in yogurt production, the validation aspect contained in the title is not explicitly mentioned. The first paragraph could be more specific about the connection between accurate pH monitoring and yogurt quality.
Authors response:
According with the reviewer suggestion, the first paragraph was completed as follows:
Yogurt is one of the preferred dairy foods by consumers as a type of multi-functional food with high nutritional value, relatively low price, and long shelf-life. Because of its huge market demand, the yogurt industrial production chain has matured, so the control of key points on the production line is required to be accurate, fast and efficient. Milk fermentation is the most important stage in yogurt manufacturing [1]. Control of milk fermentation process end point in industrial yogurt production is very important because its relationship with product quality. If the pH is high, fermentation is incomplete, resulting in yogurt with an undesirable taste and texture. If the pH is too low, the yogurt may become too sour, and the texture may separate and release whey. Also, yogurt final pH in industrial production plants require values lower to 4.6 due to food safety reasons.
- The introduction relies heavily on the limitations of traditional methods. Although it is important to focus earlier on the proposed optical technology. In the second paragraph, introduce the concept of NIR backscattering and its potential benefits.
Authors response:
We add the following new paragraph with a brief information about NIR sensor technology
A fiber optic sensor technology used to measure light dispersion has been demonstrated to be a convenient inline and nondestructive method for monitoring milk coagulation. Light from a light-emitting diode (LED) is transferred to the milk through an optic fibre, and the light backscattered (LB) from the milk is transmitted through an adjacent fibre to an optical detector. The LB signal contains information about aggregation of casein micelles and gel assembly during rennet coagulation [4], [5], [6].
- Explicitly state the connection between the validation of NIR technology and its application in yogurt production. In this context, mention the model of Arango (2005).
Authors response:
With the aim of clarify the validation aspect contained in the title and its connection with the previous study of Arango et al., the following paragraphs were modified:
Arango et al. (2020a) [3] evaluated the suitability of an optical sensor as a new method for the inline control of yoghurt fermentation. Tests were conducted with three fat and three inulin concentrations and fermentation was monitored simultaneously using an inline light backscatter sensor, pH meters and a rheometer. A mathematical model that correlates the near infrared light backscatter ratio with the pH, at real time during milk fermentation, was developed. The model was calibrated and successfully validated at the different experimental conditions.
The objective of this study was to validate the feasibility of estimating the pH of milk during the yogurt making process by a NIR light backscatter sensor under different fermentation temperatures and milk protein concentrations, using the mathematical model proposed by Arango (2020a) [3], because in this study only fat and inulin concentration were evaluated and its necessary to validate if the method and model works properly under other production conditions.
- Materials and Methods – Consider adding a short introduction to this section that summarizes the entire experimental approach.
Authors response:
We included the suggested short introduction to materials and methods section:
An experiment was designed in which yogurts were made from milk with different protein contents and with two fermentation temperatures. In each test the variation of the pH and the NIR light backscatter ratio (R) were measured, in order to obtain and validate models that transform data of R into pH measurements.
- In Section 2.2, explain why the same batch of milk was used throughout the experiment. In Section 2.5.1, you might consider mentioning the reasons for choosing 880 nm as the wavelength for NIR light.
Authors response:
In section 2.2 we explain why the same batch of milk was used throughout the experiment. As follows:
The same batch of milk was used for the whole experiment, in order to minimize the variability associated to uncontrolled milk composition or pretreatment.
At the end of first paragraph of section 2.5.1 we explain the reason for choosing 880 nm as the wavelength for NIR light as follows:
The wavelength of 880 nm has been used repeatedly in several works where the NIR light scattering sensor has been used to evaluate enzymatic and acid coagulation of milk, because it is the one that best responds to micellar aggregation processes [7], [8].
- Results and discussion – In Section 3.1, consider consolidating results and discussions. Report the observed influence of temperature and protein concentration on model performance without going into too much detail about specific values. Similarly, Sections 3.2 and 3.3 combine results and discussion, focusing on the key findings. Highlight the improvement in model performance through voltage gain adjustment and optimal conditions (3.5% protein, 43°C).
Authors response:
We agree with reviewer’s opinion and for that reason we modified the second paragraph in the section 3.1 of results and discussion:
It was evident from Table 2 that model calibration and validation were greatly affected by temperature. For protein concentration of 3.5%, increasing fermentation temperature from 43 to 46 °C resulted in a decrease of the model performance indicators. The accuracy of the pH prediction model is indicted by high value of R2, and small values of both SEP and CV [15]. Also, the observed effect of fermentation temperature on model calibration and validation for protein concentration of 4.0% was quite similar to that of 3.5% protein [16]. So as the temperature raised, the results showed that the fitting accuracy was lower. On the other hand, it was observed that increasing the protein concentration worsened the model adjustment, although the result is less clear than the effect of increasing the temperature.
Also, in section 3.2 we modified the second paragraph as follows:
According to the results shown in Tables 2 and 3, the model with adjusted initial voltage gain yielded better predictions than those of vat #2 (without adjustment). According to the inference of section 3.1.1, when protein remained the same, the higher the temperature, the lower the fitting accuracy of the prediction model. Similarly, with the voltage gain adjusted, when the protein concentration was 3.5%, the accuracy of the prediction model was reduced when the temperature increased from 43 °C to 46 °C. This same effect also applied to 4.0% protein concentration. On the other hand, similar values of R2 and SEP were obtained at 43 °C for both protein levels. This result also happened at 46 ºC. The best R2, SEP and CV values of validation were obtained at a combination of 43 °C and 3.5% protein concentration.
Finally, the last part of the third paragraph in section 3.3 was modified:
The effect of temperature and protein on the prediction coefficients was similar to those already discussed in sections 3.1 and 3.2. As in previous cases (vats #1 and #2), the fitting of the established prediction model was optimal under conditions of 3.5% of protein concentration and a fermentation temperature of 43 °C.
- Describe the key features of Figures 5, 6, and 7 in the text, emphasizing the trends and influence of temperature and protein concentration.
Authors response:
According with the reviewer suggestion, we added the following new paragraph in line 460 of the modified manuscript:
As observed in Figures 5, 6 and 7, it can be concluded that in all cases the pH pre-diction model showed a better fit at lower protein levels and fermentation temperature. The adjustment in the voltage gain allowed to improve the pH prediction at both low and high protein levels and temperature, even for the pH data obtained discontinuously with an external electrode.
- Conclusions – Quantify the improvement achieved by voltage gain adjustment. Note the limitations of the model at higher fermentation temperatures. Mention potential areas for further research. Simplify some sentences for better readability.
Authors response:
Conclusion were modified as follows:
The results showed that the model for inline prediction of pH during milk fermentation using light backscatter data was successfully validated at different milk protein concentration and fermentation temperatures. The best model fitting for pH prediction was obtained in treatments with 3.5% protein and 43 ºC fermentation temperature. Increasing fermentation temperature from 43 to 46 ºC lowered the model fit-ting accuracy, however, the R2 and SEP values were not affected with increasing milk protein concentration. Adjusting initial voltages improved the pH prediction model fitting, increasing R2 from 0.989 to 0.998, for treatments with 3.5% protein at 43 ºC. The proposed optical in-line and non-destructive method was feasible for inline pH monitoring of milk fermentation, avoiding traditional manual pH measurement. Further validation of the method is necessary to improve the fitting of pH prediction at high protein levels or fermentation temperatures. It should also be validated under other yogurt production conditions not yet studied.
- Abstract – The summary can be shortened by removing unnecessary details. Emphasize the advantages of the proposed method compared to traditional approaches.
Authors response:
Abstract was shortened and improved according to reviewer suggestions, as follows:
Current systems that allow inline pH control in fermented dairy industry have drawbacks, such as protein adhesion on the non-glass pH probes, measurement distortion, frequent recalibration needs, and sensitivity to extreme pH conditions encountered during clean in place operations. Therefore, the objective of this study was to validate the feasibility of estimating the pH of milk during the yogurt making process by a NIR light backscatter sensor measuring under different fermentation temperatures and milk protein concentrations, using a mathematical model that correlates the light scatter signal with pH. Three replications of the experiment with 2 protein concentrations (3.5 and 4.0%), and 2 fermentation temperatures (43 and 46 °C) were used to validate this inline pH prediction model. Continuous and discontinuous measurements of pH were collected as a reference during fer-mentation, simultaneously to the light backscatter data acquisition. Also, the effect of adjusting the initial voltage gain of the light scatter device on the accuracy of the pH prediction model was evaluated. Temperature and initial voltage were the main factors affecting the fitting accuracy of the model. The adjustment of the initial voltage gain improved the pH prediction model fitting. The model has been successfully validated for both continuous and discontinuous measurements of pH, with SEP values < 0.09 pH units and CV < 1.78%. The proposed optical, in-line, and non-destructive method was feasible for inline pH monitoring of milk fermentation, avoiding traditional manual pH measurement.
Reviewer 2 Report
Comments and Suggestions for Authors
I read and appreciated the work. In the attached file are my suggestions.
Regards

The quality of the English is good.
Author Response
Thank you for your comments.
All comments were approach as suggested.
Please see attached manuscript.

Reviewer 3 Report
Comments and Suggestions for Authors
Manuscript titled “Validation of an optical technology for the determination of pH in milk during yogurt manufacture” reports a series of experiments aimed at developing a NIR-based method to estimate dairy pH during yogurt production, in order to avoid direct contact with the samples. There are some comments and suggestions for the authors:
1. In lines 39-40, “Yogurt fermentation is the most important stage…”, would it be more appropriate to say that “Milk fermentation is the most important stage…”?
2. Lines 51-63 mention some general data about the proposed method. Could the authors please extend this section? Only some generalities are stated, but going into more detail could be beneficial to better understand and justify your work. A diagram of the process could also support this section.
3. Lines 67-68 mention two temperatures and two protein concentrations, can you please comment to the reader why were these specific values chosen?
4. Line 76 mentions that “voltage correction will be explained in detail later”. Please consider directing the reader to a specific section, for example, “voltage correction is explained in section X”.
5. Section 2.2 states that milk was bought from a supermarket; is the milk available in supermarkets the same as that used for industrial yogurt production? Or is there some additional pre-treatment that may be applied to milk available to consumers but not to milk for yogurt production (or vice versa)?
6. Line 256 state that findings are consistent with the works of Arango and Lee and Lucey. Please consider referencing these authors on the previous paragraph, instead of starting the new paragraph with these references.
7. Although your work is interesting and has potential, no method is perfect and has its limitations. Please consider stating what limitations this may have, and perhaps what can be improved on in future works.
Author Response
- In lines 39-40, “Yogurt fermentation is the most important stage…”, would it be more appropriate to say that “Milk fermentation is the most important stage…”?
Authors response:
The reviewer is right, the suggested change was made as follows:
Milk fermentation is the most important stage in yogurt manufacturing [1].
- Lines 51-63 mention some general data about the proposed method. Could the authors please extend this section? Only some generalities are stated, but going into more detail could be beneficial to better understand and justify your work. A diagram of the process could also support this section.
Authors response:
According with a similar suggestion of reviewer 1, we added the following new paragraph in the introduction section with a brief information about NIR sensor technology
A fiber optic sensor technology used to measure light dispersion has been demonstrated to be a convenient inline and nondestructive method for monitoring milk coagulation. Light from a light-emitting diode (LED) is transferred to the milk through a fibre, and the light backscattered (LB) from the milk is transmitted through an adjacent fibre to an optical detector. The LB signal contains information about aggregation of casein micelles and gel assembly during rennet coagulation [4], [5], [6].
- Lines 67-68 mention two temperatures and two protein concentrations, can you please comment to the reader why were these specific values chosen?
Authors response:
According with the reviewer suggestion, we added the following information in line 93 of the modified paper:
These values were chosen because they are commonly used in the dairy industry.
- Line 76 mentions that “voltage correction will be explained in detail later”. Please consider directing the reader to a specific section, for example, “voltage correction is explained in section X”.
Authors response:
The following sentence was added in line 102 of the new manuscript:
The mentioned voltage correction is explained in detail in section 2.5.2.
- Section 2.2 states that milk was bought from a supermarket; is the milk available in supermarkets the same as that used for industrial yogurt production? Or is there some additional pre-treatment that may be applied to milk available to consumers but not to milk for yogurt production (or vice versa)?
Authors response:
Milk for yogurt production is the same as that commercially available as UHT milk. But for yogurt processing its necessary to make a strong thermal treatment to milk with the aim of denaturalizing whey proteins. This process was made at 90 ºC for 5 min, as it was explained in section 2.2. Also, for yogurt milk, solids must be increased by adding milk powder.
- Line 256 state that findings are consistent with the works of Arango and Lee and Lucey. Please consider referencing these authors on the previous paragraph, instead of starting the new paragraph with these references.
Authors response:
The change asked by the reviewer was made
- Although your work is interesting and has potential, no method is perfect and has its limitations. Please consider stating what limitations this may have, and perhaps what can be improved on in future works.
Authors response:
The observations made by the reviewer were included in the conclusions of the paper, as follows:
The results showed that the model for inline prediction of pH during milk fermentation using light backscatter data was successfully validated at different milk protein concentration and fermentation temperatures. The best model fitting for pH prediction was obtained in treatments with 3.5% protein and 43 ºC fermentation temperature. Increasing fermentation temperature from 43 to 46 ºC lowered the model fit-ting accuracy, however, the R2 and SEP values were not affected with increasing milk protein concentration. Adjusting initial voltages improved the pH prediction model fitting, increasing R2 from 0.989 to 0.998, for treatments with 3.5% protein at 43 ºC. The proposed optical, in-line, and non-destructive method was feasible for inline pH monitoring of milk fermentation, avoiding traditional manual pH measurement. Further validation of the method is necessary to improve the fitting of pH prediction at high protein levels or fermentation temperatures. It should also be validated under other yogurt production conditions not yet studied.
Round 2
Reviewer 3 Report
Comments and Suggestions for Authors
Manuscript titled “Validation of an optical technology for the determination of pH in milk during yogurt manufacture” reports a series of experiments aimed at developing a NIR-based method to estimate dairy pH during yogurt production, in order to avoid direct contact with the samples. The present version of the document was modified, according to comments and suggestions made during an initial revision, those made by the present reviewer include:
1. Confirming the phrasing of a sentence in the introduction (“Yogurt fermentation is the most important stage…”). The authors corrected the sentence.
2. In the introduction, adding more information regarding the proposed method on which the authors are basing their work. Additional information was added to this section.
3. Justifying the authors’ choice of temperatures and protein concentrations. The authors state that these are common in the dairy industry.
4. Directing the reader to a specific section regarding voltage correction. The authors now direct the reader to section 2.5.2.
5. Clarifying if there are differences between supermarket milk and the one used to produce yogurt. The authors have clarified that a thermal treatment is required for yogurt production, as described in section 2.2.
6. Moving two references (Arango and Lee and Lucey) to the previous paragraph, instead of starting the new paragraph with them. The paragraph was edited.
7. Stating some limitations and potential improvements to the authors’ work. The authors have now mentioned what could be done in future works in the conclusion.
According to the changes made by the authors, it is apparent that they adequately considered and addressed this reviewer’s comments and suggestions. There are no additional ones for the present version of the document.